# Multi-layered cement-hydrogel composite with high toughness, low thermal conductivity, and self-healing capability

Yuan Chen[1], Yangzezhi Zheng[2], Yang Zhou ®[1] ✉, Wei Zhang ®[3] ✉, Weihuan Li[1], Wei She[1], Jiaping Liu[1] & Changwen Miao[1]

The inherent quasi-brittleness of cement-based materials, due to the disorder of their hydration products and pore structures, present significant challenges for directional matrix toughening. In this work, a rigid layered skeleton of cement slurry was prepared using a simplified ice-template method, and subsequently flexible polyvinyl alcohol hydrogel was introduced into the unidirectional pores between neighboring cement platelets, resulting in the formation of a multi-layered cement-based composite. A toughness improvement of over 175 times is achieved by the implantation of such hard-soft alternatively layered microstructure. The toughening mechanism is the stretching of hydrogels at the nano-scale and deflections of micro-cracks at the interfaces, which avoid stress concentration and dissipate huge energy. Furthermore, this cement-hydrogel composite also exhibits a low thermal conductivity (around 1/10 of normal cement) and density, high specific strength and self-healing properties, which can be used in thermal insulation, seismic high-rise buildings and long-span bridges.

Cement-based materials have been used by humans nearly since the dawn of civilization, and they remain the most widely used building materials in the world, with the global consumption exceeding 4 billion tons in 2022[1]. However, the inherent defects of cement-based materials significantly affect their performance and service life, especially the brittleness and low toughness, which is mainly caused by the disordered and random distribution of their hydration products (primarily calcium silicate hydrate, C-S-H) and the pore structures[2].

Cement-based materials can be toughened by incorporating reinforcing materials into the matrix[3–9], which helps to bridge cracks and enhance the material's resistance to deformation and fracture, or modify the properties of the cement slurry itself to improve its ductility and toughness[10–16]. However, this toughening method can only lead to a moderate improvement in mechanical properties, the resulting increase in ductility and toughness is typically limited to no more than twofold. Moreover, this method could have a negative

impact on the cohesiveness of the slurry preparation process, leading to the reduction of compressive strength[17–19].

Nature has inspired a promising approach to synchronously strengthen and toughen inorganic composites[20,21]. For example, the nacre in mollusk shells comprise of 95% (mass proportion) hard $CaCO_3$ crystals (aragonite) and 5% soft organics, mostly protein and chitin, possess exceptional strength and toughness due the orderly arrangement of hard (aragonite layer) and soft (protein layer) components[22–26]. The aragonite serves as the load bearing and reinforcing phase, while the organic materials act as the energy dissipation medium. This strategy of ordered arrangement of hard and soft phases has been employed to replicate the biological features into artificial structural materials, increasing the toughness of multiple inorganic matrixes[24,27,28]. Picker et al.[29] fabricated a meso-crystalline C-S-H by the construction of highly aligned C-S-H nanoplatelets interspaced with a polymeric binder. The flexural strength of this meso-crystalline C-S-H

[1]Jiangsu Key Laboratory of Construction Materials, School of Materials Science and Engineering, Southeast University, Nanjing 211189, China. [2]School of Transportation, Southeast University, Nanjing 211189, China. [3]Jiangsu Key Laboratory of Advanced Metallic Materials, School of Materials Science and Engineering, Southeast University, Nanjing 211189, China. ✉e-mail: tomaszy@seu.edu.cn; w69zhang@seu.edu.cn

(at the length scale of hundreds of micrometers) was 40-100 times larger than that of C-S-H at a macro-scale, even approaching the flexural strength of nacre. However, this method is limited at microscopic scales, which cannot be scaled up to meet engineering applications.

The ice-template method is an effective approach to address the size limitations of bio-inspired materials and has been extensively used in the preparation of multi-layered structures across different materials[30–34]. For cement-based materials, Amanmyrat et al.[35] utilized freeze-casting process to produce porous cement film scaffolds for water microfiltration and ultrafiltration processes. Wang et al.[36] developed a cement material with a unidirectional porous structure through ice-template technology and the self-hardening properties of cement. However, the current research on ice-templated cement-based materials has shown that they have advantages in hydrophobicity, thermal conductivity and other functional requirements, but the porous cement matrix is not conducive to the improvement of mechanical properties. It is necessary but difficult to select appropriate soft materials filling the cement matrix and form a bionic structure that can enhance the strength and toughness. Polymer-based hydrogels are ideal candidates for the soft fillers since they are intrinsic ductile[37–39]. Moreover, their hydrophilic groups can stably adsorb on the surface of C-S-H and other hydration products to form high mechanical performance cement-polymer composites[40]. Meanwhile, hydrogels consist of large amount of water, which may support the self-repairing and self-healing of cementitious materials[41,42]. Therefore, we believe that by constructing a cement matrix skeleton with a unidirectional pore structure using the ice-template method and filling it with a soft hydrogel phase, we can mimic the biological structure of shell nacre and achieve enhanced toughness in the resulting cement-hydrogel composite.

In this work, we construct a layered cement skeleton with unidirectional pores by ice-template method, in which in-situ hydration was used instead of freeze-drying to effectively simplify the procedure. The ordered microstructures can be retained without freeze-drying treatments since hydration reactions take place automatically during the thawing process. Subsequently, low-viscosity PVA solutions are filled into the unidirectional pores by negative pressure impregnation, and two cycles of freezing-thawing are implemented to in situ fabricate

the PVA hydrogels, as shown in Fig. 1. By elaborating such an ordered microstructure of highly aligned cement pastes interspaced with PVA hydrogels, we realize a toughness increase of over 175 times in the cementitious matrix, which is the greatest improvement in the toughness of cement-based materials that has been reported[43–48]. Based on molecular dynamics simulations and finite element analysis, the toughening mechanism of hydrogel preferentially stretching deformation and inducing crack deflection in the middle of the cement laminate was further revealed. In the meanwhile, the multifunctional performance of this cement-hydrogel composite material are also investigated and compared to traditional cement-based materials, including a density and thermal conductivity of 1.44 g/cm³ and 0.21 W/m K, respectively, reduced by about 40% and 80%, as well as self-healing properties to bridge microcracks.

## Results
### Structure and performance

As shown in Fig. 2, the cement paste with a layered structure in order at multi-scales can be fabricated by an ice-template method. At a larger length-scale (Fig. 2a, d), the X-Ray computed tomography (XCT) and energy dispersive spectroscopy (EDS) mapping results indicate that the cement paste exhibits a well-layered configuration in a long-range order, while directional pores are present in between neighboring cement paste platelets. At a smaller length-scale (Fig. 2b), it is suggested that PVA hydrogels can be effectively filled in those unidirectional pores, after the suction of PVA solutions and 2-3 cycles of freezing-thawing. At a micro-length-scale (Fig. 2c), the scanning electron microscope (SEM) images show PVA hydrogels in the interlayers form polymer networks, binding neighboring cement paste lamellae. Furthermore, PVA hydrogels are also coated on the surface of cement lamellae in a membranous manner, which can fill the micro-pores and micro-cracks in the cement paste and effectively improve the interface bonding. It should be noted that no obvious damage is observed on the cement matrix after 2-3 freezing-thawing cycles, as shown in Fig. 2a, b.

The mechanical properties of the pure cement paste, ice-templated cement paste, and cement paste-PVA hydrogel composite

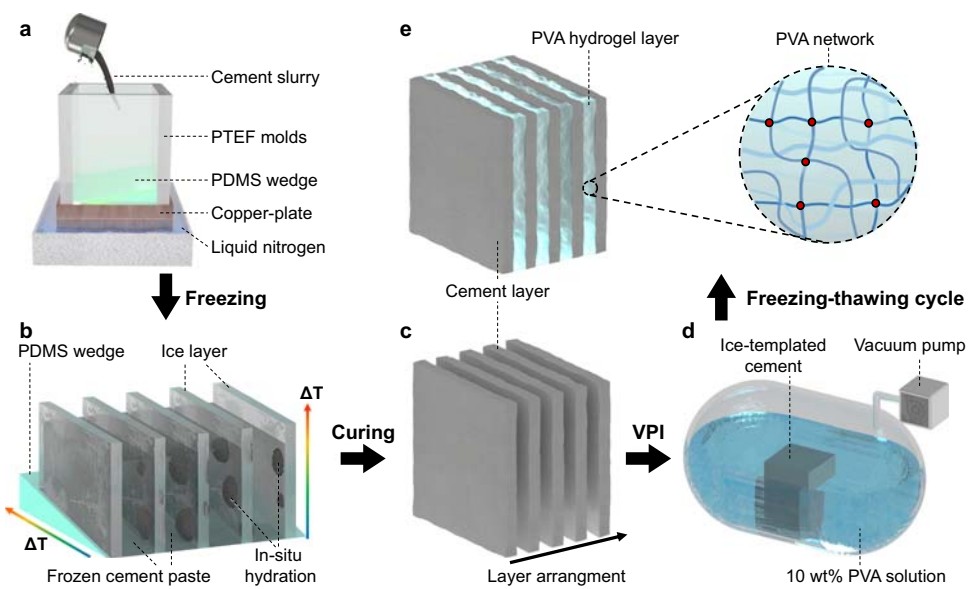

**Fig. 1 | Schematic illustrations of the fabrication process of the cement-hydrogel composite. a** The mixing of cement slurries in the mold with internal dimensions of 50 × 50 × 50 mm. **b** Cement slurries solidify into ice layers by a bidirectional freezing gradient both vertically and horizontally. Cement particles are squeezed between ice layers and slowly in-situ hydrate. **c** Cement particles hydrate into order layers during a thawing and curing period with a thickness of 10–100 μm. **d** The PVA solution was filled into the pores between neighboring cement sheets by VPI (vacuum pressure impregnating). **e** PVA hydrogels were formed between the cement lamellae after 2-3 freezing-thawing cycles.

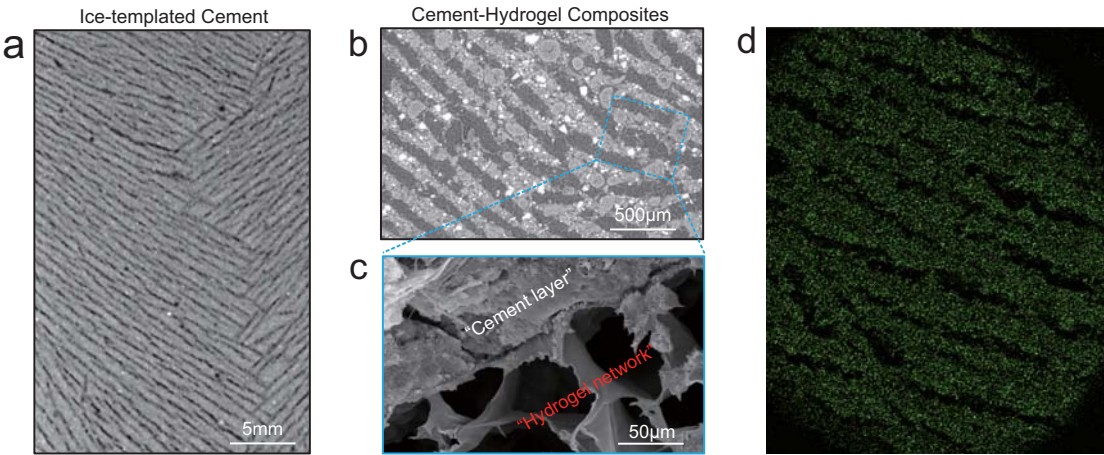

**Fig. 2 | Microstructural images of cement-hydrogel composite. a** An ice-templated cement by XCT. **b** A cement-hydrogel composite by XCT. **c** A cement-hydrogel composite by SEM. **d** A EDS mapping pattern of Si in the cement-hydrogel composite.

**Fig. 3 | Mechanical properties of the cement paste, ice-templated cement, and cement-hydrogel composite. a** Stress−strain curves during three-point flexural tests. **b** Flexural strength and flexural toughness. **c** Toughness comparisons between the cement-hydrogel composite and other materials[4–9,43,50–52].

are illustrated in Fig. 3a, respectively. According to the three-point bending tests (along the direction perpendicular to the cement lamellae), the flexural strength and toughness of cement paste are both improved approach two times (from $2.65 \pm 0.27$ MPa, $8 \pm 1.3$ kJ/m³ to $4.73 \pm 0.34$ MPa, $19 \pm 2.1$ kJ/m³) by an ice-template mixing method.

Besides, after the PVA hydrogel suction, the ductility of the cement-based composite is dramatically increased, and the flexural toughness experiences an enhancement by about 2 orders of magnitude, from $19 \pm 2.1$ kJ/m³ to $1407 \pm 153$ kJ/m³. As is known to all, ordinary cementitious materials are quasi-brittle with low toughness. However, the ice-

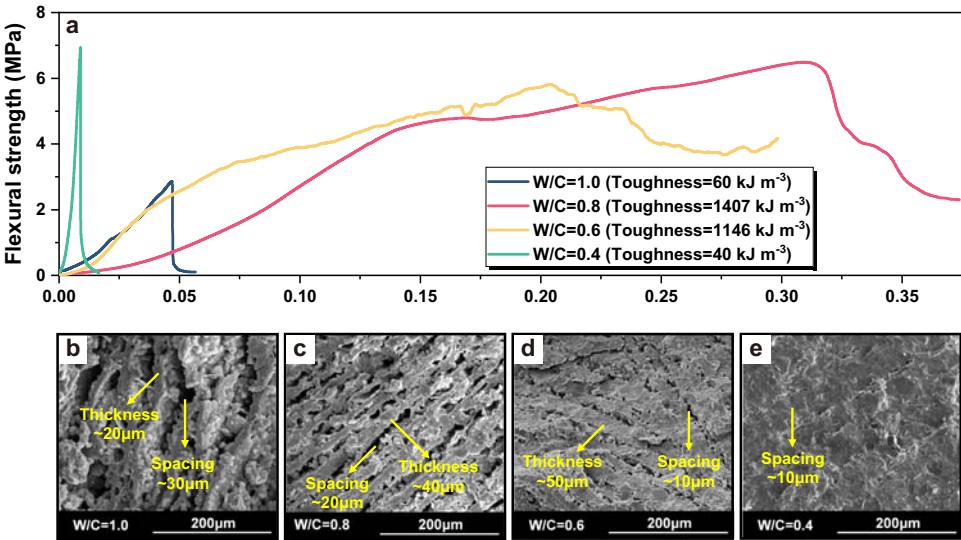

**Fig. 4 | The influence of W/C on the mechanical properties and microstructure of the cement-hydrogel composite. a** Mechanical properties of cement-hydrogel composite at different W/C ratios. SEM images of Cement-Hydrogel at (**b**) W/C = 1 (**c**) W/C = 0.8 (**d**) W/C = 0.6 (**e**) W/C = 0.4.

templating method optimizes the pore structure of cement pastes, resulting in increased flexural strength along a specific direction. Based on this, the incorporation of flexible materials into the directional pores of rigid cement matrix constitutes a lamellae structure, with hard and soft phases alternately aligned, which obtains extremely high toughness. Furthermore, a strain-hardening period even appears in the stress-strain curve of the cement paste-PVA hydrogel composite, which increases the ultimate strain to over 0.35. It should be noted that two major cracks appear in the fracture pattern of the composite, differing from either pure cement pastes (one major crack) or other high-ductility cement-based materials, such as engineered cementitious composites (ECC, multi-cracks). It indicates a distinct toughening mechanism. In ECC, the added PVA fibers can bridge the micro-cracks to avoid the rapid propagation and convergence of one major crack. The orientation of PVA fibers is random, and the microstructure of ECC can be considered homogenous, which leads to a multi-cracking failure mode[49]. However, this cement-hydrogel composite has a layered microstructure, where PVA hydrogels within neighboring cement lamellaes can prevent or deflect the cracks on the interfaces due to their low elastic modulus and high cohesiveness. Therefore, the final fracture mode of this composite is as similar as that of nacre[24,25], with a limited number of major cracks. More details with regards to the toughening mechanism can be referred to the finite element analysis in Section Interfaces and Mechanisms.

As shown in Fig. 3c, the toughness of this cement-hydrogel composite surpasses various kinds of cementitious materials, including fiber-reinforced concrete and polymer modification cement-based materials, which are two commonly-used toughening measures[50,51]. In the meanwhile, the density of the cement-hydrogel composite is significantly lower than any other cementitious materials, further intensifing the merits on the specific toughness. In comparison with the hydrogel component alone, the cement-hydrogel composite not only increases the toughness, but also significantly improves the mechanical strength. Pan Hao et al.[43] fabricated a biomimetic cement-based material by a hot-pressing method based on the brick-mortar structure and obtained a considerable toughness enhancement. However, the toughness is still approximately one order of magnitude lower than that of the cement-hydrogel composite, which may be attributed to the arbitrary packing of the cement particles induced by hot pressing. This suggests that a unified orientation of the cement matrix may make toughening more efficient. Although the toughness of cement-hydrogel composites is lower than those of metals, their specific toughness still outweighs in consideration of the much lower density[52].

The influence of water-cement ratio (W/C) on the mechanical properties of cement-hydrogel composites is illustrated in Fig. 4. The toughness of the cement-hydrogel composite is highly dependent on the W/C of cement pastes. As shown in Fig. 4a, the composite exhibits high toughness at the W/C = 0.6 and 0.8, while the enhancement efficiency sharply decreases as the W/C = 0.4 or 1.0. This can be explained by the difference in the filling efficiency of PVA hydrogels. The SEM images (Fig. 4b–e) indicate that as the W/C decreases, the thickness of each cement paste lamellae gradually increases, which in turn limits the interlayer space of directional pores. As W/C = 0.4, the thickness of the cement lamellae is over 80 μm, while the interlayer pore region is hardly observed and filled with large amounts of hydration products. Thus, it becomes difficult for PVA solutions to smoothly suction and hydrogels to densely fill the pores, resulting in low filling efficiency of PVA hydrogels and little toughness enhancement of the composite. However, if the W/C is too high, i.e. W/C = 1.0, the thickness of the interlayer pores is even larger than one of cement lamellae, which causes a much lower strength of the composite. Although the filling efficiency of PVA hydrogels is adequate, an improper proportion of the soft and hard phases still leads to a toughening failure.

## Interfaces and mechanisms

The performance of a composite largely depends on the properties of the interfaces between its components. In the case of the cement-hydrogel composite, the interfaces between each cement paste and the PVA hydrogel layer play a crucial role in enhancing the toughness of the composite. The interface bonding strength between the cement paste and PVA hydrogel is illustrated by a tensile test, as shown in Fig. 5a–c. It suggests that during the interface separation process, the PVA hydrogel tightly adheres to the surface of the cement paste and exhibits a high tensile ductility. In addition, the interface bonding strength is almost equivalent to the tensile strength of PVA hydrogel itself, indicating a high affinity between the interface. The Fourier transform infrared spectrometer (FTIR) test results of pure cement paste, PVA hydrogel, and cement-hydrogel interface are illustrated in Fig. 5d, respectively. The characteristic peak at around 2900 cm$^{-1}$ represents the vibration of C-H groups in PVA hydrogels[38,43,53], which is also present in the patterns of cement-hydrogel interface. It suggests that PVA hydrogels can penetrate the surface of the cement paste and interact with the hydration products, which is in good agreement with

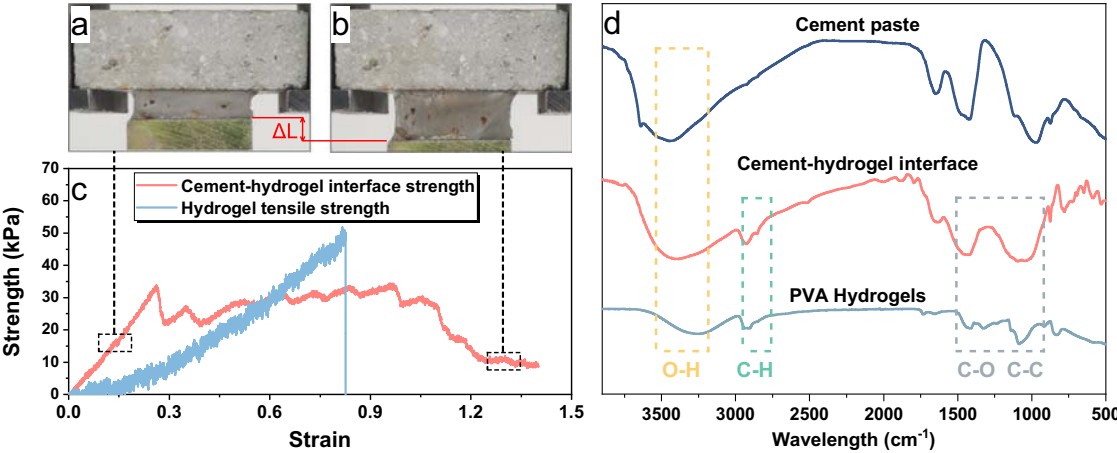

**Fig. 5 | Experimental characterizations of the cement-hydrogel interface bonding. a** Initial state of tensile cement-hydrogel interface separation process. **b** Final state of tensile cement-hydrogel interface separation process. **c** A comparison between stress-strain curve during a tensile cement-hydrogel interface separation process. **d** FTIR patterns of the cement paste, PVA hydrogels, and cement-hydrogel interface.

the interface micro-films observed in SEM images. The high interface compatibility between the cement paste and PVA hydrogel may account for the huge toughness enhancement of the composite.

Molecular dynamics is employed to investigate the interface interactions between the cement paste and PVA hydrogel at the nano-length scale. To simplify the simulation, calcium silicate hydrates, the main hydration product of Portland cement, are used to represent the cement paste matrix. The static simulation results, including interaction energy, snapshots, radial distribution function (RDF), and time correlation function (TCF), are shown in Fig. 6a–d, while the dynamic snapshots during a uniaxial tensile simulation are illustrated in Fig. 6e. The interface interaction energy between C-S-H and PVA hydrogel at dry and saturated state is −575 kcal/mol and −295 kcal/mol, respectively. It indicates a high interface affinity at both states, which is consistent with the above experimental results. This high interface affinity can be explained by three types of atomic coordination. One is the strong attractions between calcium ions on the surface of C-S-H and oxygen atoms from the hydroxyl groups of PVA. The RDF patterns show that calcium and oxygen can form atomic pairs at a distance of around 2.7 Å (Fig. 6b), which is confirmed by the interface snapshots (Fig. 6d). The TCF values of Ca-O coordination oscillate around 1 as the simulation proceeds, indicating this type of atomic pair is quite stable. The other two types of atomic coordination are H bonds formed between C-S-H and PVA. As shown in Fig. 6d, C-S-H can donate hydrogen atoms in the surface hydroxyl groups to the oxygen atoms of PVA, forming one type of H bond. Similarly, the surface non-bridging oxygen atoms of C-S-H may also accept hydrogen atoms from the hydroxyl groups of PVA, which forms another type of H bond. The RDF results suggest the bond length of these two types H bonds is both around 1.8 Å. The TCF values of these H bonds gradually decrease, implying the occurrence of bond breakages during the simulation process. It should be noted that although H bonds is not as stable as Ca-O coordination, there are large numbers of donors and acceptors of H bonds on the interface, which constitutes a huge H bonds network and provides numerous opportunities for bond formations and reorganizations. Therefore, along with Ca-O coordination, H bonds also contribute to the strong adsorption of PVA on the surface of C-S-H (see Fig. 6b–d).

In accordance with the abovementioned interface bonding experiments, a uniaxial tensile test along the direction perpendicular to the interface plane on the C-S-H/PVA hydrogel composite is simulated by molecular dynamics. With the increase of the tensile strain, the network structure of PVA hydrogels in the interlayer is continuously stretched, the intertwined PVA molecule chains is gradually separated from each other to undertake the tensile strain and prevent the fracture of the composite. Despite some PVA molecules are ruptured during the

stretching process, the remaining PVA long chains still bridge the upper and lower C-S-H matrix due to the significant interface affinity (even when the tensile strain reaches 0.8 Å/Å). The high flexibility of the PVA hydrogels and high interface affinity between C-S-H and PVA conjointly determine the huge ductility and toughness of the composite.

The finite element method (FEM) simulation results are shown in Fig. 7. When cement paste structure without ice-templating or hydrogels is subjected to three-point bending loads, the maximum tensile stresses and cracks would occur at the mid-span, where the bending moment is the greatest. As the loading increases, the stress is always concentrated at the tip of the crack, which leads to the rapid crack propagation. Eventually, the whole structure fractures almost instantaneously. However, in the cement-hydrogel composite, the hydrogel layer effectively mitigates stress concentration and crack growth. The stress and crack development process in the layeredcement-hydrogel composite can be divided into four steps. At step one, as the maximum tensile stress is lower than the cracking stress of the cement layer, the overall structural stress distribution is more dispersed compared to the conventional cement. In the meanwhile, as the modulus of the hydrogel is much smaller than that of the cement, it makes a band distribution of stresses between cement layers. At step two, similar to the cement paste structure, cracks begin to appear in the lowermost cement layer across the middle. However, the neighboring hydrogel layer limit the development of cracks, instead, multiple cracks begin to appear at both sides where stress concentration is present. At step three, with the gradual increase of loadings, a crack at the right side is the first to propagate significantly and gradually expands to the middle, while apparent deflections occur near the crack tip during the progressive interfacial separation of the cement-hydrogel composite. Eventually, at step four, the whole structure is penetrated by the cracks and fails. The bond failure process between the cement and the hydrogel is also accompanied by micro-crack sprouting and interface sliding, which synergistically promote crack deflection and bridging. In conclusion, the FEM simulation results prove that the hydrogel layer effectively limits the crack expansion behavior and also enhances the overall structural ductility.

## Multi-functionalities

In addition to the toughening effect, the cement-hydrogel composite also exhibits multi-functionality. Figure. 8a presents the density, specific compressive strength, and thermal conductivity coefficients of the pure cement, ice-templated cement, and cement-hydrogel composite, respectively, along the direction perpendicular to the lamellae plane at the same W/C = 0.6. Compared with the pure cement, the other two materials that experience an ice-template

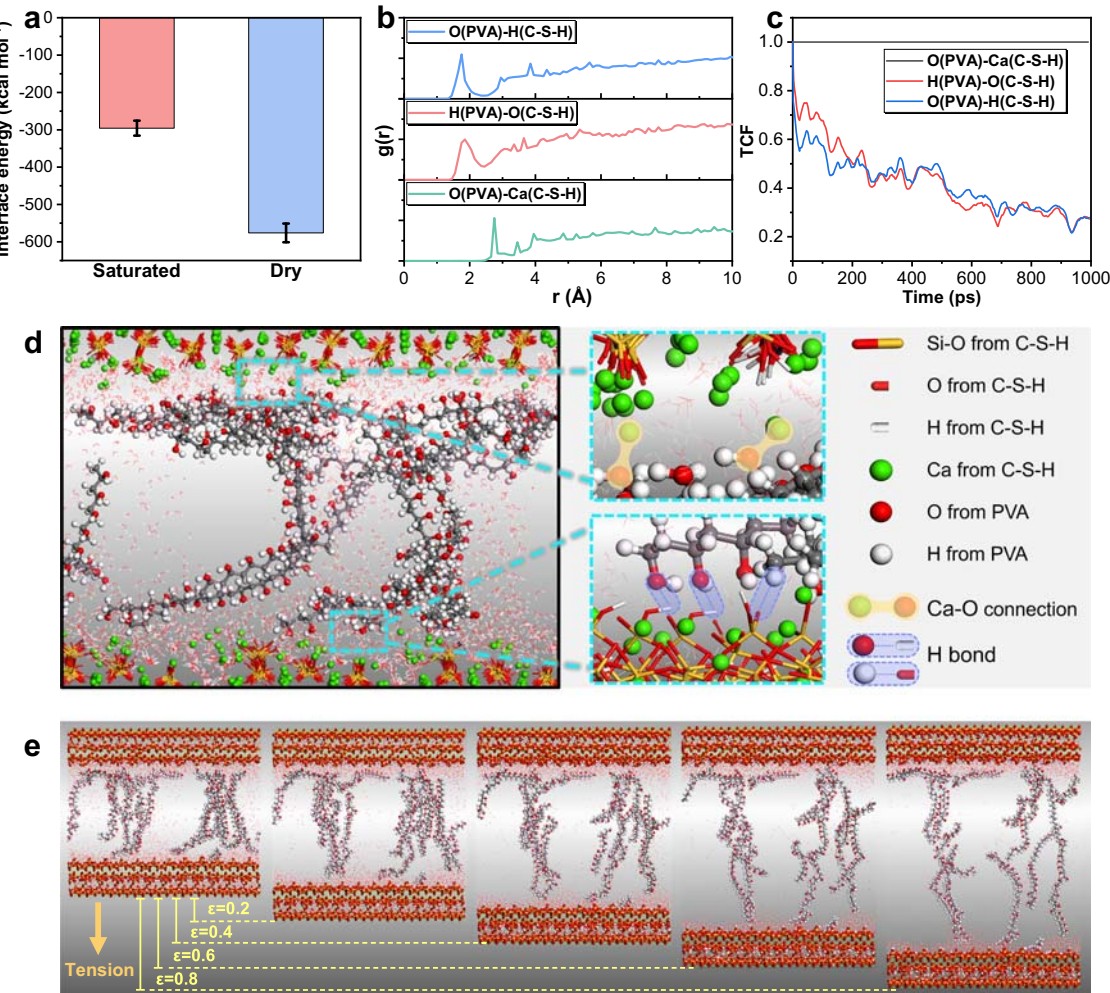

**Fig. 6 | Molecular dynamics characterizations of the cement-hydrogel interface bonding (For C-S-H, green balls denote calcium ions, yellow and red sticks denote silicate tetrahedra, red and white sticks denote hydroxyl groups; for PVA, grey, red, and white balls denote carbon, oxygen, and hydrogen atoms, respectively). a** Interface interaction energies between C-S-H and PVA hydrogel at saturated and dry states. **b** RDF profiles of different interfacial connections on the C-S-H/PVA interface. **c** TCF profiles of different interfacial connections on the C-S-H/PVA interface. **d** A snapshot of Ca-O connections and H bonds on the C-S-H/PVA interface. **e** Snapshots of the C-S-H/PVA composite tensioned at the strain of 0, 0.2, 0.4, 0.6 and 0.8.

process have significantly lower densities, resulting in a specific compressive strength that is still greater than that of pure cement paste along the direction normal to the lamellae plane, despite the presence of orientated pores due to the ice-template mixing. It should be noted that it is the weakest direction of this composite with respect to compressive strength. In addition, it is also reported that the ice-templated cement has a significantly higher specific strength along other compressive directions[36]. Furthermore, Fig. 8a shows the ice-template method also effectively decreases the thermal conductivity coefficient due to the uniform orientation of the pore structure, which can slow down the heat transfer. In contrast with the ice-templated cement, the cement-hydrogel composite exhibits an even lower thermal conductivity coefficient. It is related to the crosslinking polymer network as shown in Fig. 2c, which partitions relatively large interlayer pores into small ones, leading to improved the thermal insulation. This multi-layered structure can decrease the thermal conductivity coefficient of pure cement paste by almost one order of magnitude, making is even close to that of aerated concrete, a commonly-used construction material with substantially lower strength[54]. The integrated function of low density, thermal insulation and high mechanical properties suggests potential applications of this cement-hydrogel composite in fields such as aerospace, military, and building energy conservation.

Furthermore, the multi-layered structure of the cement-hydrogel composite offers opportunities for the self-healing and self-repairing of cementitious matrix. As the internal relative humidity decreases, excessive water can be released from the PVA hydrogels and react with unhydrated cement particles or pozzolanic materials to heal the micro-cracks in the cement lamellae. Although superabsorbent polymers have already been applied in cementitious materials for self-healing, the direct mixing of polymers and cements usually leads to a significant loss of strength[55]. Here, the lamellae structure of this cement-hydrogel composite enables self-healing without compromising its mechanical properties. In contrast with the previously published wood-like cement[36], the cement-hydrogel composite fabricated in this work not only retains its multifunctional properties, including high specific compressive strength and effective thermal insulation at the transverse profile, but also significantly improves the toughness of the matrix,. In the meanwhile, the self-healing possibilities supported by hydrogels are beneficial to the durability of cement-based materials, which may promote the applications in the construction fields that require long-term sealing and are difficult to repair, such as the disposal of nuclear wastes.

A multi-layered cement-based composite has been fabricated by a two-step method. Firstly, a simplified ice-template mixing is utilized to produce an ordered platelet-like skeleton of cement pastes, with uni-directional pores aligned in the interlayer. Subsequently, PVA

**a**

**Cement paste**

Stage 1: Initial bending

S11 (MPa)

+3.00
+2.50
0.00
-2.50
-3.00
-4.50
-6.00

Stage 2: Single crack generation

Stage 3: Crack penetration

Stage 4: Destruction

**b**

Stage 1: Stress laminar distribution

**Ice-template cement**

Hydrogel layer hinders the extension of cracks

Stage 2: Multi-crack generation

S11 (MPa)

+15.00
+7.50
0.00
-7.50
-15.00
-22.50
-30.00

Main crack direction

Stage 3: Crack penetration

Stage 4: Destruction

**Fig. 7 | Simulation results of the finite element method. a** The stress distribution and crack propagation of the cement paste. **b** The stress distribution and crack propagation of the cement-hydrogel composite.

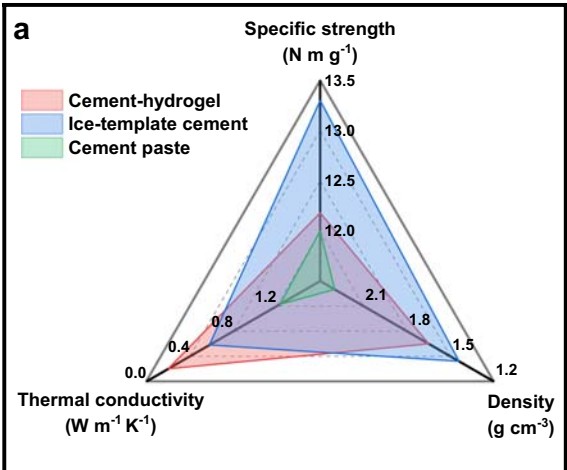

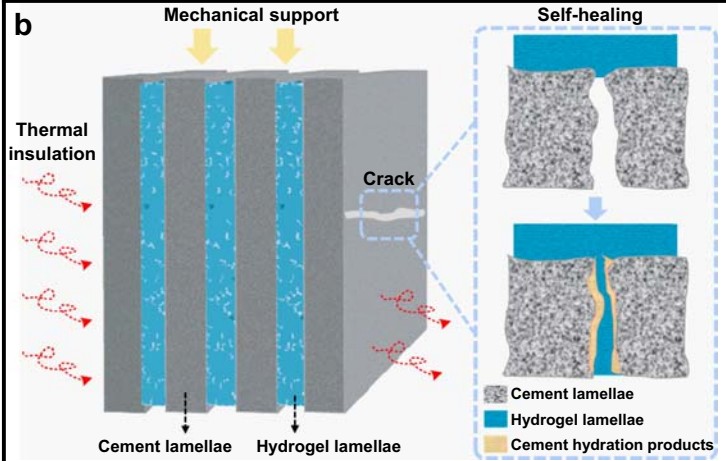

**Fig. 8 | Schematic illustrations of a multi-functionality of the cement-hydrogel composite. a** Comparisons of the density, specific compressive strength along the direction normal to the lamellae plane, and thermal conductivity of pure cement paste, ice-templated cement, cement-hydrogel composite. **b** The integration of thermal insulation, water impermeability, self-healing with high mechanical properties.

hydrogels are in situ filled into the unidirectional pores. By constructing such a layered microstructure of highly aligned cement pastes interspaced with PVA hydrogels, we realize a toughness improvement of over 175 times as high as pure cement paste, 1-2 orders of magnitude higher than that of the most commonly used fiber toughening measures. In the meanwhile, the flexural strength of the pure cement is also improved by over 2 times.

Furthermore, the toughness enhancement mechanism of the cement-hydrogel composite has also been interpreted. The bonding strength of cement-hydrogel interface is close to the tensile strength of PVA hydrogel itself, which indicates super high affinities between the hard and soft phase. The excellent interface compatibility results from the stable adsorption of PVA molecules on the cement matrix. Molecular dynamics simulations suggest as the tensile stress is applied, the PVA hydrogel can be fully stretched and deformed before the interface de-bonding, which significantly improves the ductility of the cement-hydrogel composite. Besides, the FEM simulations implys that during the progressive interface failure process, this multi-layered structure leads to distinct micro-crack deflections near the crack tip, also accompanied by interface sliding and platelets pull-out. Those mechanisms can effectively avoid the stress concentration and dissipate huge energy, which explains the large ductility increase in the flexural stress-strain curves.

In addition to the toughening and strengthening effect, the cement-hydrogel composite is further accompanied with multi-functionalities, including low density (around 2/3 of the pure cement paste), thermal insulation (one order of magnitude lower than that of the pure cement), and self-healing. With a future development trend of construction industrialization, the assembly line production in factories can make the large-scale preparation of this cement-based composite possible and reduce the costs. Therefore, we believe the composite may promote a significant revolution of building materials, which can be widely applied in seismic high-rise or energy-conservation buildings, and long-span bridges.

## Methods

### Materials
Ordinary Portland cement (P.O. I 42.5) was offered by United Cement Co. Polyvinyl-alcohol (PVA, analytical grade, molecular weight 80,000–90,000) was purchased from Aladdin reagent. Polydimethylsiloxane (PDMS) was purchased from Dow Corning.

The use of PVA was to prepare a 10 wt% polyvinyl alcohol solution, with PVA added to deionized water, stirred in a water bath at 105 °C for two hours. The use of PDMS was to prepare a wedge with a 15° inclination on the copper plate of the mold.

### Preparation methods
Firstly, a wood-like cement paste skeleton was fabricated by a simplified ice-template method. Ordinary Portland cement powder and deionized water were mixed at different water-cement ratios (0.4, 0.6, 0.8, 1.0, and 1.5) by stirring for 3 min. Then the cement slurries were cast into square tetrafluoroethylene molds with internal dimensions of 50 × 50 × 50 mm attached onto a copper plate. The copper plate at the bottom of the mold was connected to a copper column immersed in liquid nitrogen for heat transfer. The PDMS wedges were placed at the bottom inside the mold, to build a bidirectional freezing gradient both vertically and horizontally[56]. After frozen, the slurries were refrigerated at 2–5 °C for 2 days to allow for slow thawing, which preserved the lamellar structure of the cement paste due to the hydration reactions between cement particles and surrounding melted ice[36]. Subsequently, the slurries were immersed in water for curing. The curing environment was 20 °C ± 2 °C and the relative humidity is above 95% RH.

The PVA was filled into the interlayer pores of the cement paste after the curing age reaches. The cement paste was placed into a vacuum oven for 2 days drying at 65 °C, and then immersed in the prepared 10 wt% polyvinyl alcohol solution. Vacuum negative pressure was employed to continuously press PVA solutions into the cement until no obvious bubbles emerged from the cement block. To facilitate the filling of hydrogels into the cement gaps, a low viscosity PVA solution (~10 mPa•s) and longer impregnation time were used. After the solution was stable, the filled cement blocks were placed in a refrigerator at −30 °C for 8 h, and then thawed at room temperature for 2 h, completing a single freezing-thawing cycle. Finally, the cement-hydrogel composite with internal pores filled with PVA hydrogels was obtained after 2-3 freezing-thawing cycles. It should be noted that a proper W/C ratio was chosen and the cement matrix had been completely dried before hydrogel filling to ensure a minimal damage induced by freezing-thawing cycles.

### Mechanical testing
All the mechanical tests were conducted at room temperature on cement-hydrogel composites that were processed into test blocks with dimensions of 30 × 10 × 8 mm. Three to five samples were prepared for testing in each group under different forming parameters, and the average test results of multiple samples were taken as experimental results. The three-point bending test was carried out using the universal testing machine CMT4503, with the maximum bearing capacity of the testing machine 300KN and the loading rate 0.5 mm/min. The bending loading was along the direction perpendicular to the cement lamellae. The stress-strain relationship during the loading process was recorded, and the flexural toughness is calculated from the stress-strain curve, defined as the area enveloped by the curve. The compressive and interfacial bonding strength tests were carried out using the universal testing machine UMT5105, with the maximum bearing capacity of the testing machine 100KN. The loading rate in the compressive test was 250 N/s. The interface bonding strength test was conducted in accordance with Chinese Standards for basic performance test of building mortar JGJ 70-2009, using a 70 × 70 × 20 mm cement block was utilized as a matrix, while PVA hydrogels placed on the surface of the cement block with a 40 × 40 mm area and a 6 mm thickness. The load rate was 5 mm/min during the test.

### Microstructure characterizations
The microstructure and the interface morphology of the cement-hydrogel composite were characterized using a field-emission scanning electron microscope (SEM FEI 3D) operated at an accelerating voltage of 20 kV. The element mapping was analyzed by scanning the sample surface 16 times using an EDAX elemental analyzer. Prior to imaging, the samples were dried and coated with a thin layer of gold. The meso-architecture of the cement lamella and hydrogels in the interlayer were detected using X-ray computed tomography (Zeiss Xradia 510 Versa). The samples were rotated by 360° with respect to the normal axis of the detector, with 1000–1400 slices of 2D projections taken. The FTIR spectra were obtained using a Nicolet iS10 spectrometer within the wavenumber range of $400 – 4000 \, cm^{-1}$.

### Thermal conductivity characterization
The thermal conductivities of samples of cement paste and cement-hydrogel composite were determined using the transient plane source method, in accordance with ISO22007-2. The samples were prepared as plates with dimensions of 20 × 20 × 4 mm, and their thermal conductivity was measured using a Hot Disk TPS 2500 S thermal conductivity meter, in both the direction perpendicular to the lamellar and parallel to the lamellar direction. The testing was conducted at a temperature of 20 °C and each sample was tested once. To ensure the reliability of the results, at least three samples were tested for each group, the average results of the multiple samples were used as the experimental data.

## Molecular dynamics simulations

In the molecular dynamics simulations, the cement paste matrix was simplified by the main hydration product, calcium silicate hydrate (C-S-H), which is a key determinant of the macro-performance of cementitious materials. Following the construction procedure in Zhou's study[11], the C-S-H model employed in this study was built according to the methodology of the "realistic model" proposed by Pellenq et al.[57], which has been widely applied into the chemical and physical behavior simulations of C-S-H[58,59]. Subsequently, the model was cleaved along the (001) plane (the center of the interlayer space), and the basal spacing was enlarged to construct a gel pore, and the final C/S is set to 1.3. Although the average C/S ratio of real C-S-H in pure Portland cement is around 1.7, the addition of high dosage supplementary binding materials (e.g. fly ash, slag or silica fume) usually decreases the value[60–63]. After constructing the two-layered calcium silicate hydrate (C-S-H) structure, the Grand Canonical Monte Carlo (GCMC) adsorption method was employed to fill PVA molecular chains and water molecules into the interlayer space of the C-S-H layer, in order to simulate a PVA hydrogel in cement layer. The density of PVA hydrogel was 1.02 g/cm³, which is consistent with the experimental results, and the molecular weight was 12,000-14,000. The resulting C-S-H/PVA/C-S-H layered structure contains approximately 28,000–32,000 atoms and possessing x, y, z dimensional dimensions of approximately $88 \times 88 \times 67$ Å³. The model construction was conducted by Materials Studio 8.0 software in the High Performance Computing Centre in Nanjing University, and the final structure is shown in Supplementary Fig. 1.

A static simulation of the C-S-H/PVA interfacial interactions and a uniaxial tensile simulation of the C-S-H/PVA interface was both carried out on the LAMMPS software. The Clay Force Field (ClayFF)[64] and Consistent Valence Force Field (CVFF)[65] were employed to simulate the energy, structure, and dynamic properties of the C-S-H and PVA model, respectively, and the force field parameters are described in Supplementary Table 1–7. The interfacial atomistic interactions were based on the average law. The combination of ClayFF and CVFF has been widely applied into the molecular dynamics simulations of cement/polymer composites, and verified to be quite suitable for the organic-inorganic interface modeling[11,37,66].

For the static simulation, a canonical ensemble (NVT) was employed at temperature 298 K, with a Nose– Hoover thermostat and a time step of 1 fs to integrate the Verlet equations of motion. An equilibration time of 3 ns was used to guarantee complete interactions between polymers and C-S-H, while another 3 ns run was performed for the production period[11]. The selected equilibrium time of 3 ns is sufficient for the system to reach an equilibrium state, as shown in Supplementary Movie 1. Furthermore, for the uniaxial tensile simulations, the interface between C-S-H and PVA models were separated at a strain rate of 0.008 ps⁻¹, along the z-axis, which was the C-S-H/PVA lamella stacking arrangement direction[40,67,68], as shown in Supplementary Movie 2. During both simulations, the trajectories of all atoms in the system were sampled every 0.1 ps for the statistical analysis of structure and dynamics, including parameters of interaction energy, radial distribution function (RDF), time correlation function (TCF).

The interaction energy reflects the interfacial binding ability between two phases (positive values denote repulsions and negative values denote attractions). To calculate the energy of each phase, the system can be divided into two regions, one corresponding to phase 1 and the other to phase 2 in LAMMPS. The energy of each phase can then be calculated using the potential energy function that describes the interactions between the atoms or molecules in the system. Once the energy of each phase is calculated, the interface interaction energy can be obtained by subtracting the energy of each phase from the total energy of the system. A larger absolute value indicates a stronger interfacial interaction.

RDF, defined as the probability of finding an atom B located at a distance r from an atom A, represents the spatial correlations between different atoms. It can be used to characterize the interfacial interactions between C-S-H and PVA, as the Eq. (1)[11] shows:

$$g_{AB}(r) = \frac{\langle N_{AB}(r) \rangle}{\rho dV(r)} \tag{1}$$

where $\langle N_{AB}(r) \rangle$ denotes the ensemble average number of atoms B located at a distance between r and r + dr away from the given atom A, $\rho$ denotes the atomic density of atoms B, $dV(r)$ is the corresponding volume of the shell between r and r + dr.

TCF, defined as Eq. (2)[11], is employed to evaluate the connection stability of atomistic pairs between C-S-H and PVA. As the simulation time passes, the TCF will be the constant 1 if all interfacial connections remain unchanged. Otherwise, TCF values will decrease as long as some unstable connections are broken. A higher degree of TCF values deviating from 1 implies an atomistic pair with lower connection stability and strength.

$$TCF = \frac{\langle \delta b_{(t)} \delta b_{(0)} \rangle}{\langle \delta b_{(0)} \delta b_{(0)} \rangle} \tag{2}$$

where $\delta b_{(t)} = b_{(t)} - \langle b \rangle$, $b_{(t)}$ is a binary operator. If a connection (e.g., Ca−O) is made, $b_{(t)}$ equals to 1, and if the connection is broken, it equals to 0. $\langle b \rangle$ denotes the average of this operator for all of the pairs and over all of the simulation times.

## Finite element method simulation

The commercial software ABAQUS was used to simulate the stress distribution and crack development in the multi-layeredcement-hydrogel composite during the flexural strength test.

The results of SEM and other microscopic experiments illustrate that the thickness of the cement layer is about 40 μm and that of the hydrogel layer is about 10 μm. However, due to the limitation of ABAQUS unit division and computational power, the structural thickness of each layer has to be increased by 10 times. The size of specimen is $30 \times 10 \times 8.4$ mm, and the span of the lower rigid bodies is 25 mm, with a cement layer thickness of 0.4 mm and a hydrogel layer thickness of 0.1 mm in an alternating arrangement. The crack fracture process is simulated using the cohesive unit of extended finite element method (XFEM), and according to the mechanical test data, three kinds of cohesive units were set up for inter-cement layer, interlayer of hydrogel and cement layer, and possible crack area. The elastic modulus and Poisson's ratios of the cement matrix were determined to be 26.8 GPa and 0.2, respectively. For the hydrogel, the corresponding values were 0.24 MPa and 0.48. The zero-thickness cohesive elements were used to simulate the fracture properties between the cement and the hydrogel. The cohesive elements obeyed the Maxs damage model and was set to be removed when the displacement reached 0.0005 mm, which was expressed as a fracture process between the cells, macroscopically, as shown in Supplementary Movie 3. Meanwhile, the conventional cement structures with the same dimensions and material properties were also modeled for comparison, as shown in Supplementary Movie 4. Crack formation is induced in conventional cement by presetting an initial crack of 1 mm.

## Reporting summary

Further information on research design is available in the Nature Portfolio Reporting Summary linked to this article.

## Data availability
The authors declare that the data supporting the findings of this study are available within the paper and its supplementary information files.

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

## Acknowledgements

Y.Z. and J.P.L. acknowledge the National Key Research and Development Project of China (Grant No: 2021YFF0500800). Y.Z. acknowledges supports from the National Natural Science Foundation of China (Grant Nos: 52250010 and 52050128). W.Z. acknowledges supports from the Fundamental Research Funds for the Central Universities (Grant No: 2242023K40027). The authors also wish to thank the High Performance Computing Center in Nanjing University.

## Author contributions

Y.Z. conceived and designed experiments. Y.C. and Y.Z.Z.Z. performed the experiment and wrote the paper. W.Z. gave the comments on manuscript modification. W.H.L. performed the data visualization and presentation. W.S. provided experimental equipment, materials and other tools. J.P.L. and C.W.M. provided supervision and leadership of the experiment. All the authors analyzed the results and provided feedback on the manuscript.

## Competing interests

The authors declare no competing interests.
