## [Peer Review File · Nature Communications]

nature portfolio

Peer Review File

Multi-layered cement-hydrogel composite with high toughness, low thermal conductivity, and self-healing capabilityReviewers' comments:

Reviewer #1 (Remarks to the Author):

In this paper, a biomimetic hydrogel cement is investigated. It is claimed that this composite material achieves 200 times the toughness of ordinary cement with a lower density and higher specific strength, which would interest the materials community. However, these claims should be supported by meticulous experiments, rigorous analysis, thorough and thoughtful discussion of the mechanisms involved, and detailed explanations for the community to understand. Unfortunately, this paper needs to include these categories at a level required for a publication such as nature communications. The following is a partial list of my comments:

1 - The provided experimental results don't seem to have any error bars. Were the experiments carried out on a single sample? This raises serious reproducibility questions.

2 - Very little detail is provided for the molecular structure of the C-S-H used. What's the C/S ratio? What software was used for GCMC calculations etc?

3 - The experiments are carried out at microscopic scales. The interfacial behavior cannot be due to the intercalation of the polymer into C-S-H layers. So how is a molecular model with polymer inside the layers justified for studying the mechanisms?

4 - How exactly is this "interface interaction energy" calculated, and how does it relate to the experimental strength? This is probably one of the vaguest parts of the paper.

5 - How is the thermal conductivity measured? No details are presented in that regard.

6 - Can the Ca-O bonding be verified with experiments (e.g., with NMR or IR)?

7 - The literature review is rather weak in this paper. There is a large amount of literature on organic-inorganic cementitious materials that can help put the study in context.

8 - There are no feasibility arguments about the composite material in the paper. What are the drawbacks of this composite in practice?

Some more minor issues:

9 - What is meant by "deflections of micro-cracks"?

Line 24: "The mechanical performances of biological materials are far beyond their artificial counterparts": not necessarily true.

Line 44: "production of cements accounts for 7~10% CO2 emissions" this seems an overestimation. Please provide a credible source for this number.

Line 54: "subversively" doesn't seem to be the right word here.

Line 106: Figure 2 is referenced before figure 1

Figure 1: what's the scale of these ? scale/dimensions are vital here.

Line 133: "optimization of pore structure of cement pastes." Where is the proof that this is the contributing factor?

Line 151: reference [40] is not by Wei She et al in the bibliography

Line 215: All abbreviations should be introduced the first time they're used (TCF, RDF etc)

Reviewer #2 (Remarks to the Author):

In this work, authors have reported a biomimetic strategy to develop cement composite with high toughness, low density, and low thermal conductivity for multiplex applications through bio-inspired directional freeze casting of the slurry of the cement followed by impregnation of the freeze-cast cement porosities or lamellae by PVA-derived hydrogels. Besides they studied the self-healing and mechanical properties of the developed composite cement paste they also conducted a molecular dynamic simulation to unravel the interface interaction between the cement and PVA hydrogels at a molecular/ nano length scale.

Unfortunately, the directional freeze casting approach has been hugely reported for the development of biomimetic e.g. nacre mimetic structural nanomaterials as well as thermal insulation materials. Just please check these two important reviews:

1. <https://doi.org/10.1002/adem.202000033>
2. [10.1002/adma.201907176](https://doi.org/10.1002/adma.201907176)

Besides, I do not see any significant improvement in the mechanical strength/ toughness as well as thermal insulation of the proposed cement! The thermal conductivity achieved by this cement composite is very huge as compared with other reported lightweight super-insulating porous materials with thermal conductivity less than $0.02 \text{ W m}^{-1} \text{ K}^{-1}$ developed by the same freeze-casting concept.

Therefore since the fabrication strategy is quite old and the developed materials have not significant improvement in their achieved key properties I do not recommend the publication of this manuscript in Nature communication, as this paper won't have a significant contribution to the field!

Response to reviewers

We really appreciate all the thoughtful comments and suggestions on our work, which are very valuable to improve the quality of our manuscript! Each suggested revision and comment, brought forward by the reviewers was accurately incorporated and considered. Below the comments of the reviewers are response point-by-point and the revisions are indicated in revised manuscript.

Reply to Reviewer 1:

In this paper, a biomimetic hydrogel cement is investigated. It is claimed that this composite material achieves 200 times the toughness of ordinary cement with a lower density and higher specific strength, which would interest the materials community. However, these claims should be supported by meticulous experiments, rigorous analysis, thorough and thoughtful discussion of the mechanisms involved, and detailed explanations for the community to understand. Unfortunately, this paper needs to include these categories at a level required for a publication such as nature communications.

Response:

Thank you very much for your comments and professional advice. These opinions help to improve academic rigor of our article. Based on your suggestion and request, we have made corrected modifications on the revised manuscript. We also appreciate your clear and detailed feedback and hope that the explanation has fully addressed all of your concerns.

1 - The provided experimental results don't seem to have any error bars. Were the experiments carried out on a single sample? This raises serious reproducibility questions.

Response:

It was our negligence that no error bar was added to the experimental results. However, we wish to clarify that our experiments were not conducted on a single sample, and we tested at least three samples in each group to ensure the reliability of our results. The most effective samples were selected for analysis.

Regarding the mechanical properties, we report the flexural strength and toughness of three different samples: pure cement paste, ice-template cement paste, and cement-hydrogel. The results are presented as mean values with standard deviation (SD) denoted by the \pm symbol. The flexural strength of pure cement paste, ice-template cement paste, and cement-hydrogel samples were found to be 2.91 ± 0.26 MPa, 4.65 ± 1.23 MPa, and 5.41 ± 1.43 MPa, respectively. Similarly, the toughness values were found to be 11.13 ± 3.13 kJ/m³, 19.3 ± 4.06 kJ/m³, and 1254 ± 108 kJ/m³, respectively.

In terms of thermal properties, we report the thermal conductivity of the same three samples. The mean values with SD are presented as follows: the thermal conductivity of pure cement paste, ice-template cement paste, and cement-hydrogel were found to be 1.1 ± 0.12 W/m K, 0.68 ± 0.14 W/m K, and 0.36 ± 0.08 W/m K, respectively.

We have revised our manuscript to include error bars in the Fig.3(b), Fig.6(a) and the quantity of samples in the Section 4.3 (Page 18, Line 418) and Section 4.5 (Page 19, Line 453).

2 - Very little detail is provided for the molecular structure of the C-S-H used. What's the C/S ratio? What software was used for GCMC calculations etc?

Response:

We understand the importance of providing complete and transparent information and we have made necessary additions to our revised manuscript. In Section 4.6 (Page 19-21, Line 458-522), we provide a detailed description of the simulation structure and methodology, including the mechanism of model construction, C/S ratio, structure size, simulation methods and parameters, simulation software, etc. We hope that these additions will address the reviewer's concerns and improve the overall quality of our manuscript.

As mentioned, the C/S ratio we used in the molecular dynamic simulation was 1.3 (Page 19, Line 465). This value is typical and commonly used in synthetic C-S-H systems and has been reported in previous research (*Compos. B. Eng.* 2019, 162, 433-444; *Phys. Chem. Chem. Phys.*, 2018, 20, 8247). We have included the relevant references in our revised manuscript (see Ref. [52-55]) to further support this choice of parameter setting.

In addition, we utilized the Grand Canonical Monte Carlo (GCMC) method for our calculations in Materials Studio 7.0 (Page 19, Line 474).

3 - The experiments are carried out at microscopic scales. The interfacial behavior cannot be due to the intercalation of the polymer into C-S-H layers. So how is a molecular model with polymer inside the layers justified for studying the mechanisms?

Response:

Our main goal of molecular dynamics simulations was to qualitatively investigate the interaction mechanism between the hydrogel and cement matrix, instead of accurately reproducing experimental data on a microscopic scale.

On the one hand, molecular dynamics simulation provides a way to reveal the interfacial bonding mechanism between C-S-H and PVA on the nano scale. The simulation results can be used as a basis for finite element simulation, which can further

explain the interfacial bonding mechanism between C-S-H and PVA. Moreover, SEM microstructure characterization can also provide support for this conclusion. The SEM image in Fig. 2c demonstrated that PVA hydrogels formed polymer networks in the interlayers, binding neighboring cement paste lamellae.

Therefore, by combining the results of molecular dynamics simulation and SEM characterization, we can gain a better understanding of the interfacial bonding mechanism between C-S-H and PVA.

4 - How exactly is this “interface interaction energy” calculated, and how does it relate to the experimental strength? This is probably one of the vaguest parts of the paper.

Response:

In Section 4.6 (Page 20, Line 495-503), we have provided a more detailed description of the calculation method of interface interaction energy.

The interaction energy reflects the interfacial binding ability between two phases (positive values denote repulsions and negative values denote attractions). To calculate the energy of each phase, the system can be divided into two regions, one corresponding to phase 1 and the other to phase 2 in LAMMPS. The energy of each phase can then be calculated using the potential energy function that describes the interactions between the atoms or molecules in the system. Once the energy of each phase is calculated, the interface interaction energy can be obtained by subtracting the energy of each phase from the total energy of the system. A larger absolute value indicates a stronger interfacial interaction.

It is important to note that the "interface interaction energy" is not directly related to strength, but rather reflects the interfacial forces between the C-S-H and the polymer layer. This approach was employed in our study to gain insight into the interfacial behavior of the cement-hydrogel composite system, as a supplement to the interface bonding experiment.

5 - How is the thermal conductivity measured? No details are presented in that regard.

Response:

We apologize for the oversight in not adequately describing the methodology for measuring thermal conductivity, and in the revised manuscript, we have made the necessary additions to the Section 4.5 (Page 19, Line 446-455) to provide a detailed description of how thermal conductivity was measured.

The thermal conductivities of samples of cement paste and cement-hydrogel composite were determined using the transient plane source method, in accordance with

ISO22007-2. The samples were prepared as plates with dimensions of 20 mm × 20 mm × 4 mm, and their thermal conductivity was measured using a Hot Disk TPS 2500S thermal conductivity meter, in both the direction perpendicular to the lamellar and parallel to the lamellar direction. The testing was conducted at a temperature of 20°C and each sample was tested once. To ensure the reliability of the results, at least three samples were tested for each group, the average results of the multiple samples were used as the experimental data.

6 - Can the Ca-O bonding be verified with experiments (e.g., with NMR or IR)?

Response:

Characterizing Ca-O bonding in cement-based composites can pose a challenge. As per the reviewer's suggestion, we attempted to use NMR to characterize Ca-O bonding in our cement-based composites. However, due to the poor signal-to-noise ratio and the lack of a clear reference method, we were unable to obtain effective results. Similarly, IR spectroscopy can also be used to study Ca-O bonding, but it is also influenced by the cement-hydrogel composite material system. Our results using FTIR spectroscopy did not enable the characterization of Ca-O bonding, as shown in Fig.5d of the manuscript.

Based on previous studies and simulation results, it has been suggested that Ca-O bonding can be observed in pure phase C-S-H composite systems using Scanning transmission X-ray microscopy (STXM), as reported in *Langmuir* 2017, 33, 14, 3404–3412 and *ACS Appl. Mater. Interfaces*, 2017, 9(46), 41014–41025. These findings support the accuracy of our simulation of Ca-O bond cooperation at the CSH and PVA interfaces. However, replicating this test is challenging as we do not have access to the synchrotron radiation source necessary to conduct the experiment.

7 - The literature review is rather weak in this paper. There is a large amount of literature on organic-inorganic cementitious materials that can help put the study in context.

Response:

As mentioned, there have been extensive studies on organic-inorganic cementitious materials, and we have revised the manuscript accordingly. While there is indeed a large body of literature on organic-inorganic cementitious materials, we have highlighted the unique contribution of our study in using the ice-templating method to create a layered structure in cement-based materials.

In the revised manuscript, we have included additional relevant literature for reference. Firstly, we have cited references [3-10] and [13-16] to introduce the common method and mechanism of incorporating organic matter and using organic-inorganic interfaces to enhance cementitious material properties, among which Ref. [8-10] and Ref.[15-16] were newly introduced. (Page 2, Line 30-33)

Secondly, we have explained our rationale for selecting hydrogels based on their affinity with the cement matrix and flexibility. To support this approach, we have cited Ref. [32-38] in the field of cement-based materials. (Page 3, Line 64)

8 - There are no feasibility arguments about the composite material in the paper. What are the drawbacks of this composite in practice?

Response:

We have added the discussion on drawbacks in Section 3, Line 373-379.

Based on the excellent mechanical properties, thermal insulation and other properties of cement-hydrogel composite materials, the application scenarios of this material include: thermal insulation, seismic building and long span bridge.

Regarding the limitations of the material, we acknowledge that the two-step preparation process (ice template and vacuum pumping) of the cement-hydrogel composite material is more complex and time-consuming than conventional cement-based materials, as illustrated in Section 4.2, Page 17, Line 392-413 and Fig.1. However, we believe that the superior properties of the composite material make it a promising candidate for certain applications that require high performance.

Some more minor issues:

9 - What is meant by “deflections of micro-cracks”?

Response:

The term "deflections of micro-cracks" refers to the phenomenon where micro-cracks are easily deflected by resistance during the initial stage of crack propagation.

In our study, we have found that the interlamellar hydrogel in the cement-hydrogel composite material system can effectively provide this resistance. This is due to the ability of hydrogels to absorb and dissipate energy, which helps to prevent the propagation of micro-cracks and improve the overall toughness of the material..

Line 24: “The mechanical performances of biological materials are far beyond their artificial counterparts”: not necessarily true.

Response:

In the revised version, we have removed this sentence and have included new sentences to emphasize the value of biomaterials research: “Nature has inspired a promising approach to synchronously strengthen and toughen inorganic composites”. (Page 2, Line 38)

Line 44: “production of cements accounts for 7~10% CO2 emissions” this seems

an overestimation. Please provide a credible source for this number.

Response:

In the revised version, we have removed this sentence.

As per the Global Carbon Budget, cement production is responsible for approximately 4% of global carbon dioxide emissions, but in China, this figure is higher and exceeds 7%. (*ESSD, 14, 4811–4900, 2022*)

Line 54: “subversively” doesn’t seem to be the right word here.

Response:

In the revised version, we have removed this word.

Line 106: Figure 2 is referenced before figure 1

Response:

In the revised version, we have rectified this error, as shown in Page 4, Line 81 and Page 5, Line 104.

Figure 1: what’s the scale of these ? scale/dimensions are vital here.

Response:

In the revised version, we have provided the scale and dimension in Fig.1, as shown in Page 4, Line 94 and Page 4, Line 98.

Line 133: “optimization of pore structure of cement pastes.” Where is the proof that this is the contributing factor?

Response:

As demonstrated in Fig. 2c, we initially transformed the disordered pore structure in the cement matrix into a directional pore structure using the ice template method. We then filled the cement pores with hydrogel, which effectively reduced porosity, eliminated harmful pores, and optimized the pore structure. This optimization ultimately led to an enhancement in the overall performance of the composite material.

Furthermore, we have also presented proof of this optimization using finite element simulations, as shown in Fig. 7. After optimizing the pore structure, the cement-based composite exhibited a deflection phenomenon of cracking, which was not observed in traditional cement-based materials without pore structure optimization.

Line 151: reference [40] is not by Wei She et al in the bibliography

Response:

In the revised version, we have rectified this error, as shown in Page 6, Line 148.

Line 215: All abbreviations should be introduced the first time they're used (TCF, RDF etc)

Response:

In the revised version, we have carefully reviewed and revised the entire manuscript to address all of your comments and suggestions. (Page 5, Line 106 and Line 112; Page 8, Line 194; Page 9, Line 214; Page 11, Line 259.)

Reply to Reviewer 2:

In this work, authors have reported a biomimetic strategy to develop cement composite with high toughness, low density, and low thermal conductivity for multiplex applications through bio-inspired directional freeze casting of the slurry of the cement followed by impregnation of the freeze-cast cement porosities or lamellae by PVA-derived hydrogels. Besides they studied the self-healing and mechanical properties of the developed composite cement paste they also conducted a molecular dynamic simulation to unravel the interface interaction between the cement and PVA hydrogels at a molecular/ nano length scale.

Response:

We appreciate your willingness to provide comments that will help us improve our work even further, and we have carefully considered your suggestions and have incorporated them into our revised manuscript. We hope that our answers have addressed your questions and met your expectations.

Unfortunately, the directional freeze casting approach has been hugely reported for the development of biomimetic e.g. nacre mimetic structural nanomaterials as well as thermal insulation materials. Just please check these two important reviews:

- 1. <https://doi.org/10.1002/adem.202000033>**
- 2. [10.1002/adma.201907176](https://doi.org/10.1002/adma.201907176)**

Response:

We agree that the directional freeze casting approach has been extensively studied in the literature, as the reviewer mentioned (*Adv. Eng. Mater.* 2020, 22, 2000033; *Adv. Mater.* 2020, 32, 1907176). While it is true that the directional freeze casting approach has been widely studied in the literature, we have made several innovative contributions to this approach that distinguish our work from previous studies.

First, we have applied the directional freeze casting approach to the field of cement-based materials, which has not been extensively explored (*Adv. Sci.* 2021, 8, 2000096; *J. Membr. Sci.* 2017, 6, 67). Second, we have improved upon this approach by

introducing a bidirectional freezing gradient, combining it with cement hydration reaction, and filling the cement matrix with hydrogel material to create a bionic structure of shell pearl layer. These innovations simplify the steps, reduce consumption, and significantly improve the mechanical and other properties of the resulting composite material.

While the ice template method has been reported in the literature, our emphasis in this paper is on demonstrating the application of this technology to the field of cement-based materials, and showing the significant improvements in properties that can be achieved. We believe that our work makes a valuable contribution to the field, and we hope that our revised manuscript will reflect this more clearly.

Besides, I do not see any significant improvement in the mechanical strength/toughness as well as thermal insulation of the proposed cement! The thermal conductivity achieved by this cement composite is very huge as compared with other reported lightweight super-insulating porous materials with thermal conductivity less than 0.02 W m⁻¹ K⁻¹ developed by the same freeze-casting concept.

Response:

As shown in Table 1 and Figure 1, the increase in toughness of our cement-hydrogel composite is the largest reported in the field of cement-based materials to the best of our knowledge. Moreover, the cement-hydrogel composites exhibit low thermal conductivity while maintaining high strength, which is a unique advantage not present in other lightweight super-insulating porous materials currently available. This makes these materials attractive for a wide range of applications, including thermal insulation, seismic building, and long-span bridge construction.

Although there may be stronger materials in the field of cement-based composites, such as fiber-reinforced composites, cement-hydrogel composites are unique in their ability to simultaneously possess low thermal conductivity and high toughness. Lightweight foamed concrete and porous aerogel materials, while exhibiting low thermal conductivity, are not suitable for use as structural materials due to their lower strength. Therefore, we consider the development of cement-hydrogel composites to be a significant breakthrough in achieving superior performance in cement-based materials.

Table 1 Comparison of mechanical and thermal conductivity properties of cement-hydrogel

Material	Thermal conductivity (W/m K)	Compression strength (MPa)	Flexural toughness (kJ/m ³)	Reference
Cement-Hydrogel composite	0.12	30.3	1407	This work
Ordinary	> 1	20-40	<160	Adv. Sci. 2021, 8,

Cement-based material				2000096; ACS Appl. Mater. Interfaces 2020, 12, 53297–53309
Graphene oxide nanocomposite	0.015	0.2	<800	Nat. Nanotechnol., 2015, 10, 277-283; Macromolecules, 2018, 51, 1696 – 1705
SiO ₂ composite aerogel	0.02-0.08	< 10	<200	Adv. Funct. Mater. 2020, 30, 2005928; J Appl Polym Sci.2020;137:e49338
Other thermally insulating material	0.015-0.2	0.1-13	10-70	Nat Commun 2012,3; ACS Nano 2022, 16, 4, 6625-6633

Figure 1 Toughness comparisons between the cement-hydrogel composite and other materials

Therefore since the fabrication strategy is quite old and the developed materials have not significant improvement in their achieved key properties I do not recommend the publication of this manuscript in Nature communication, as this paper won't have a significant contribution to the field!

Response:

Although the directional freeze casting approach has been previously reported in the literature, we have made several significant improvements to the preparation process, including in situ combination of the ice template method and cement hydration reaction, negative pressure impregnation filled polyvinyl alcohol, etc. These modifications have enabled us to achieve a more efficient and effective preparation process, and to create

a novel composite material with unique properties.

Moreover, the focus of this paper is to demonstrate the innovative application of this technology in the field of cement-based materials, and to show that significant performance improvements, such as increased toughness and reduced thermal conductivity, can be achieved. We believe that our work has made a valuable contribution to this area and we have made revisions to better highlight this contribution. We hope that our revisions reflect the importance of our findings.

Finally, We would like to take this opportunity to thank you for all your time involved and this great opportunity for us to improve the manuscript. We hope you will find this revised version satisfactory.

REVIEWER COMMENTS

Reviewer #2 (Remarks to the Author):

Unfortunately, I am not still convinced by the responses to my first revision comments and strongly believe that the concept of the paper is old, and the cement composite properties have not been sufficiently improved to become suitable for publication in the Nature Communication.

Reviewer #3 (Remarks to the Author):

Brittleness has always been a big problem for cement-based materials since the advent of cements. In this work, the toughness of the cement paste was improved by over 200 times, by means of the design and construction of a biomimetic structure, which is really interesting and striking. The cement was transformed from a conventional disordered microstructure to a hard-soft alternatively layered one, by a combination method of ice-templating and PVA vacuum filling. It changes the traditional understanding of the disordered structure of cement and concrete, and may help improve the future processing method of high mechanical performance cement-based materials. Overall, the manuscript is well-written with convincing experimental results and detailed mechanism interpretations (the MD and FEM simulations). The authors have satisfactorily addressed the reviewers technical concerns. It could be qualified for publication in Nature Communications if the following issues are properly addressed.

1. The composite was subjected to freezing and thawing cycles during the fabrication process to ensure the formation of PVA hydrogels. However, it is well-known that freezing and thawing cycles may damage the cement matrix. Have the authors noticed this damage? Please explain.
2. PVA hydrogels within the interlayer of the cement paste may gradually lose moisture during a drying period. Can this high toughness be maintained even if the hydrogel is completely dried?
3. The authors claim that "To the best of our knowledge, it is the greatest improvement in the toughness of cement-based materials that has been reported", however, they have not provided any supporting reference for this claim. Therefore, it is recommended to revise the manuscript to include a reference that supports this statement. This will strengthen the credibility of their claim and help readers to better understand the significance of their work.
4. Could the authors explain why cement slurries need to be solidified into ice layers by a bidirectional freezing gradient both vertically and horizontally.
5. The authors are advised to revise Figure 6 to include the relevant information related to 6-cement paste or remove the reference to 6-cement paste from the figure legend if it is not necessary.
6. The authors use molecular dynamics simulations to investigate the interface interaction, with a combined force field. Can this force field accurately describe the organic/inorganic interface? Why authors choose the C-S-H with a C/S=1.3, since the average C/S in cement should be around 1.7? What is density and molecular weight of PVA in the interlayer region? Is it in agreement with the experiments?
7. For the static MD simulation, an equilibration time of 3 ns was used, is it enough to reach an equilibration state since the model is relatively large (28,000-32,000 atoms)? For the uniaxial tensile simulations, why do the authors choose the strain rate 0.008 ps⁻¹? Furthermore, the authors calculate the interaction energy between C-S-H and PVA, can this value be correlated to the interface binding strength as shown in Fig.5?
8. Some minor errors:
Line 90: "g/cm3" should be "g/cm³"
Line 309: "a commonly-used construction materials" should be "a commonly-used construction

material”

Line 466: “in in” should be “in”.

Line 528: “experimental” should be “experiments”.

Reviewer #4 (Remarks to the Author):

This manuscript presents a novel approach for preparing a bionic cement-based material using the ice template method, which exhibits remarkable mechanical properties and extremely low thermal conductivity. This is a revolutionary advancement for modern cement-based materials and concrete, as it leads to anti-cracking, thermal insulation, and self-healing properties. It seems the authors have satisfactorily addressed the technical concerns from previous reviewers. Therefore, I recommend it could be accepted after the following concerns are well addressed.

1 - In Fig.2, it seems the width of interlayer pores is lower than 100 micrometers after the in situ hydration of cements. Can the PVA solution smoothly penetrate such small pores and fill any gaps? How to control the filling efficiency of PVA hydrogels?

2 - In Fig.3a, there appears two main cracks as the cement-hydrogel composite is about to the failure, which is different from the typical phenomenon of cement-based materials (usually one main crack). It is also not the multi-cracking mode as ECC, can authors explain this?

3 - The fabrication process of the cement-hydrogel composite is complex, with an ice-template process, in situ hydration and then vacuum pumping, and 2~3 freezing-thawing cycles. Can this be applied in practical large-scale construction scenarios? What should be improved in the fabrication process to make the future applications easier and more convenient?

4 - In section 4.7, some key parameters are missing in the Finite Element Method Simulations. Except for the geometry parameters of the model, the authors should also provide the mechanical modulus of the cement and hydrogel, respectively. Furthermore, how do they deal with the interface cohesiveness in the model of the cement-hydrogel composite?

5 - The conclusion section is too lengthy and may be difficult for readers to quickly understand the main points. To improve clarity, please revise the conclusion section by including only the most significant conclusions achieved in the investigation. Any detailed results or discussions should be moved to the results and discussion section.

Title: A Biomimetic Cement-Hydrogel Composite with Super High Toughness and Multi-Functionalities

Response to reviewers

Reviewer #2

Unfortunately, I am not still convinced by the responses to my first revision comments and strongly believe that the concept of the paper is old, and the cement composite properties have not been sufficiently improved to become suitable for publication in the Nature Communication.

Response:

We are disappointed that our revised manuscript still cannot be recognized by reviewer #2. However, we have tried our best to demonstrate the novelty and significant improvements in our previous response.

Although the directional freeze casting approach has been extensively studied in recent years, few studies have focused on its application in the field of cement-based materials. We innovatively adopted the design concept of composite materials and prepared the bionic composite structure of cement-hydrogel using the ice template method for the first time.

In terms of the performance, the toughness of cement-hydrogel composite is significantly improved, to the best of our knowledge, it is the largest increase that has been reported in cement-based materials. In addition, they are unique in possessing both low thermal conductivity and superior mechanical properties. Thus, we consider the development of cement-hydrogel composites to be a major breakthrough in terms of achieving excellent performance.

Reviewer #3

Brittleness has always been a big problem for cement-based materials since the advent of cements. In this work, the toughness of the cement paste was improved by over 200 times, by means of the design and construction of a biomimetic structure, which is really interesting and striking. The cement was transformed from a conventional disordered microstructure to a hard-soft alternatively layered one, by a combination method of ice-templating and PVA vacuum filling. It changes the traditional understanding of the disordered structure of cement and concrete, and may help improve the future processing method of high mechanical performance cement-based materials. Overall, the manuscript is well-written with convincing experimental results and detailed mechanism interpretations (the MD and FEM simulations). The authors have satisfactorily addressed the reviewer's technical concerns. It could be qualified for publication in Nature Communications if the following issues are properly addressed.

Response:

We appreciate the comments from the reviewer #3, and we have tried our best to address reviewers' suggestions in the following section and incorporated them into the revised manuscript.

1 - The composite was subjected to freezing and thawing cycles during the fabrication process to ensure the formation of PVA hydrogels. However, it is well-known that freezing and thawing cycles may damage the cement matrix. Have the authors noticed this damage? Please explain.

Response:

Typically, cement may suffer damage only after undergoing hundreds of freeze-thaw cycles, so the damage caused by 2-3 freeze-thaw cycles in this paper is considered negligible. As shown in Fig.2b-d, the SEM images of cement-hydrogel sample also demonstrate that cement matrix remains well-preserved after freeze-thaw cycles.

It is important to note that the cement matrix was thoroughly dried before filling it with hydrogel, which reduces the impact of water solidification expansion during the freeze-thaw cycles. Moreover, significant damage during the freeze-thaw cycles of the cement-hydrogel sample only occurred with high water-cement ratios. This is because a thin cement layer under high water-cement ratios cannot withstand the internal stress caused by volume expansion. However, the water-cement ratio investigated in this study ranged from 0.6 to 0.8, resulting in a relatively thick and dense cement layer. Therefore, under this water-cement ratio, the damage from the freeze-thaw cycles is not apparent.

To make it clear, we have added the following discussion in the revised manuscript, Page 5, line 116-117 and Page 17, line 416-418.

Revision:

It should be noted that no obvious damage is observed on the cement matrix after 2-3 freezing-thawing cycles, as shown in Fig.2a-b.

It should be noted that a proper W/C ratio was chosen and the cement matrix had been completely dried before hydrogel filling to ensure a minimal damage induced by freezing-thawing cycles.

2 - PVA hydrogels within the interlayer of the cement paste may gradually lose moisture during a drying period. Can this high toughness be maintained even if the hydrogel is completely dried?

Response:

Although the toughness of cement-hydrogel composites may decrease as the water content of the hydrogel decreases, the final toughness of the completely dried sample remains over one order of magnitude higher than that of pure cement. This is due to

water penetrating the cement matrix for hydration as hydrogel water content decreases, which strengthens the interface between the hydrogel and cement matrix and also reinforces the cement matrix.

3 - The authors claim that “To the best of our knowledge, it is the greatest improvement in the toughness of cement-based materials that has been reported”, however, they have not provided any supporting reference for this claim. Therefore, it is recommended to revise the manuscript to include a reference that supports this statement. This will strengthen the credibility of their claim and help readers to better understand the significance of their work.

Response:

As stated in the introduction (Page 2, line 30-37) and demonstrated in Fig.3(c), numerous studies have been conducted on improving the toughness of cement-based materials. However, the improvement in toughness achieved in those studies was lower than that achieved in this experiment.

In the revised manuscript, Page 3, line 83-84, we have supplemented references that support our perspective, which included a discussion of both traditional toughening methods and bionic toughening methods.

Revision:

To the best of our knowledge, it is the greatest improvement in the toughness of cement-based materials that has been reported⁴³⁻⁴⁸.

Reference:

- [43] Pan H, et al. *ACS Applied Materials & Interfaces*, 2020, 12, 53297-53309.
- [44] Peng, H., Ge, Y., Cai, C. S., Zhang, Y. & Liu, Z. *Construction and Building Materials*, 2019, 194, 102-109.
- [45] Safiuddin M, Yakhlaf M, Soudki K A. *Construction and Building Materials*, 2018, 164, 477-488.
- [46] Zhao L, et al. *Composites Part B: Engineering*, 2017, 113, 308-316.
- [47] Hu Y, Luo D N, Li P H, Li Q B, Sun G Q. *Construction and Building Materials*, 2014, 70, 332-338.
- [48] Chen X L, Lim S K J, Liang Y N, Zhang L Y, Hu X. *ACS Sustainable Chemistry & Engineering*, 2019, 7, 105-110.

4 - Could the authors explain why cement slurries need to be solidified into ice layers by a bidirectional freezing gradient both vertically and horizontally.

Response:

Vertical and horizontal bidirectional freezing method is beneficial to generating ice sheets with an ordered arrangement (*Adv Eng Mater.* 2020, 22, 2000033; *Adv. Mater.* 2020, 32, 1907176; *Adv. Sci.* 2021, 8, 2000096). Subsequently, melting and hydration can result in a cement matrix with a layered structure, which can optimize the

performance of cement-based materials by regulating their pore structure. However, structures formed under unidirectional freezing gradients are mostly characterized by random distributed pore structures, which cannot lead to the significant improvement of cement-based materials' mechanical performance.

5 - The authors are advised to revise Figure 6 to include the relevant information related to 6-cement paste or remove the reference to 6-cement paste from the figure legend if it is not necessary.

Response:

We wonder if the reviewer refers to Fig.3c instead of Fig.6. In the revised manuscript, Page 7, we have added the missing information about cement paste.

Revision:

Figure 3 Mechanical properties of the cement paste, ice-templated cement, and cement-hydrogel composite. (a) Stress–strain curves during three-point flexural tests. (b) Flexural strength and flexural toughness. (c) Toughness comparisons between the cement-hydrogel composite and other materials^{4-9,43,50-52}.

Reference:

[4] Banthia, N. & Sappakittipakorn, M. *Cem. Concr. Res.* 2007, 37, 1366–1372.
 [5] Wang, R., Wang, P. M. & Li, X. G. *Cem. Concr. Res.* 2005, 35, 900–906.
 [6] Krystek, M. et al. *Adv. Sci.* 2019, 6, 1801195.
 [7] Dimov, D. et al. *Adv. Funct. Mater.* 2018, 28, 1705183.
 [8] Liang, R., Liu, Q., Hou, D., Li, Z. & Sun, G. *Cem. Concr. Res.* 2022, 152, 106675.
 [9] Sun, G., Liang, R., Zhang, J., Li, Z. & Weng, L. T. *Cem. Concr. Compos.* 2017, 78,

57–62.

[43] Pan H, et al. *ACS Applied Materials & Interfaces*, 2020, 12, 53297-53309.

[50] Li, J. J., Wan, C. J., Niu, J. G., Wu, L. F. & Wu, Y. C. *Constr. Build. Mater.* 2017, 131, 449–458.

[51] Fernández-Ruiz, M. A., Gil-Martín, L. M., Carbonell-Márquez, J. F. & Hernández-Montes, E. *Constr. Build. Mater.* 2018, 173, 49–57.

[52] Studart, A. R. *Adv. Mater.* 2012, 24, 5024–5044.

6 - The authors use molecular dynamics simulations to investigate the interface interaction, with a combined force field. Can this force field accurately describe the organic/inorganic interface? Why authors choose the C-S-H with a C/S=1.3, since the average C/S in cement should be around 1.7? What is density and molecular weight of PVA in the interlayer region? Is it in agreement with the experiments?

Response:

In Section 4.6, Page 19, line 463-482, we provide a detailed explanation of the modeling and principles underlying our molecular dynamics simulation. We employed the combination of ClayFF and CVFF, which has been extensively used in previous studies on cement/polymer composites and has proven to be effective in modeling organic-inorganic interfaces, as reported in *ACS Appl. Mater. Interfaces*. 2017, 9, 41014–41025; *Compos. B. Eng.* 2019, 177, 107421; *Compos. B. Eng.* 2020, 194, 108036.

The C/S ratio was set to 1.3 in our simulation. Although the actual C/S ratio of C-S-H in pure Portland cement ranges from 0.6 to 2.1, the addition of high dosage supplementary binding materials (e.g. fly ash, slag or silica fume) usually decreases the value, as demonstrated in *Cem. Concr. Res.* 2010, 40, 971–983 and *Compos. B. Eng.* 2019, 162, 433–444.

The density of PVA between the layers was set to 1.02g/cm³, consistent with the experimental results. The molecular weight was approximately 12760, smaller than the range of 89000-98000 used in the experiment. However, this was primarily due to the scale limitations of the molecular dynamic simulation, and the high molecular weight PVA was not reproducible in the lattice.

7 - For the static MD simulation, an equilibration time of 3 ns was used, is it enough to reach an equilibration state since the model is relatively large (28,000-32,000 atoms)? For the uniaxial tensile simulations, why do the authors choose the strain rate 0.008 ps⁻¹? Furthermore, the authors calculate the interaction energy between C-S-H and PVA, can this value be correlated to the interface binding strength as shown in Fig.5?

Response:

The selected equilibrium time of 3ns is sufficient for the system to reach an equilibrium state, as shown in the video of the Supporting Information “MD CSH-PVA static simulation”.

Although the deformation behavior of cement-based composites depends on the strain rate, a similar response can be observed from stress-strain curves for each composite material (*J. Mater. Civ. Eng.*, 2022, 34, 04022127). At the strain rate of 0.008ps^{-1} , it is beneficial to unravel the detailed configuration evolution and interface bonding mechanism of materials, as indicated by the following reference (*Phys. Chem. Chem. Phys.*, 2018, 20, 8247; *Mater. Chem. Phys.*, 2014, 146, 503-511).

It is important to note that the "interface interaction energy" is not directly related to strength, but rather reflects the interfacial forces between the C-S-H and the polymer layer. This approach was employed in our study to gain insight into the interfacial behavior of the cement-hydrogel composite system, as a supplement to the interface bonding experiment.

We have supplemented the following discussion at Page 20, line 491-503 in the experimental section 4.6.

Revision:

For the static simulation, a canonical ensemble (NVT) was employed at temperature 298 K, with a Nose-Hoover thermostat and a time step of 1 fs to integrate the Verlet equations of motion. An equilibration time of 3 ns was used to guarantee complete interactions between polymers and C-S-H, while another 3 ns run was performed for the production period. The selected equilibrium time of 3ns is sufficient for the system to reach an equilibrium state, as shown in the video of the Supporting Information "MD CSH-PVA static simulation". Furthermore, for the uniaxial tensile simulations, the interface between C-S-H and PVA models were separated at a strain rate of 0.008ps^{-1} , along the z-axis, which was the C-S-H/PVA lamella stacking arrangement direction^{40,67,68}. During both simulations, the trajectories of all atoms in the system were sampled every 0.1 ps for the statistical analysis of structure and dynamics, including parameters of interaction energy, radial distribution function (RDF), time correlation function (TCF).

Reference:

[40] Zhou Y, Hou, D S, Geng G Q, et al. *Physical Chemistry Chemical Physics*, 2018, 20, 8247-8266.

[67] Sun, D W. Lan, M Z, Wang, Z M, et al. *Journal of Materials in Civil Engineering*, 2022, 34, 04022127.

[68] Hou, D., Zhu, Y., Lu, Y. & Li, Z. *Materials Chemistry and Physics*, 2014, 146, 503–511.

8 - Some minor errors:

Line 90: "g/cm3" should be "g/cm³".

We have revised corresponding content, as shown in Page 4, Line 90.

Revision:

including a density and thermal conductivity of 1.44 g/cm^3 and 0.21 W/m K ,

Line 309: “a commonly-used construction materials” should be “a commonly-used construction material”

We have revised corresponding content, as shown in Page 14, Line 320.

Revision:

a commonly-used construction material with substantially lower strength,

Line 466: “in in” should be “in”.

We have revised corresponding content, as shown in Page 19, Line 471.

Revision:

Although the average C/S ratio of real C-S-H in pure Portland cement is around 1.7.

Line 528: “experimental” should be “experiments”

We have revised corresponding content, as shown in Page 21, Line 537.

Revision:

The results of SEM and other microscopic experiments illustrate that the thickness of the cement layer is about $40 \mu\text{m}$ and that of the hydrogel layer is about $10 \mu\text{m}$.

Reviewer #4

This manuscript presents a novel approach for preparing a bionic cement-based material using the ice template method, which exhibits remarkable mechanical properties and extremely low thermal conductivity. This is a revolutionary advancement for modern cement-based materials and concrete, as it leads to anti-cracking, thermal insulation, and self-healing properties. It seems the authors have satisfactorily addressed the technical concerns from previous reviewers. Therefore, I recommend it could be accepted after the following concerns are well addressed.

Response:

We thank the reviewer for carefully reading the manuscript and for providing positive and very valuable feedback.

1 - In Fig.2, it seems the width of interlayer pores is lower than 100 micrometers after the in situ hydration of cements. Can the PVA solution smoothly penetrate such small pores and fill any gaps? How to control the filling efficiency of PVA hydrogels?

Response:

As shown in Fig.2c, a network distribution of polyvinyl alcohol can be observed between the cement sheets, suggesting that the polyvinyl alcohol solution can effectively fill small gaps between them.

The filling efficiency of PVA solution is mainly affected by the viscosity or fluidity of the solution. We used a lower viscosity PVA (~10 mPa s) solution for better fluidity and easier filling of the cement layer gaps during vacuum negative pressure impregnation. In addition, the vacuum negative pressure environment also plays a crucial role, as lower pressure and longer impregnation times facilitate the filling of hydrogels.

To make it clear, we have supplemented the following discussion at Page 17, line 410-412 in the experimental section 4.2.

Revision:

To facilitate the filling of hydrogels into the cement gaps, a low viscosity PVA solution (~10 mPa·s) and longer impregnation time were used.

2 - In Fig.3a, there appears two main cracks as the cement-hydrogel composite is about to the failure, which is different from the typical phenomenon of cement-based materials (usually one main crack). It is also not the multi-cracking mode as ECC, can authors explain this?

Response:

Ordinary cement has poor deformation ability, and when the internal stress exceeds the tensile strength, it results in cracks that rapidly expand and evolve into a main crack. In contrast, ECC relies on a crack-bridging mechanism through fibers to inhibit the rapid propagation of cracks, ultimately leading to a multi-crack cracking mechanism. However, the toughening mechanism of the cement-hydrogel composite in this paper is distinct from that of ECC. According to the finite element simulation presented in this paper, the high deformation ability of the interlayer hydrogel plays a crucial role in evenly distributing stress and suppressing the rapid propagation of cracks. Hydrogels can prevent or deflect the cracks on the interfaces due to their low elastic modulus and high cohesiveness. Therefore, the final fracture mode of this composite is as similar as that of nacre, with a limited number of major cracks.

To make it clear, we have added the following discussion at Page 6, line 138-151.

Revision:

It should be noted that two major cracks appear in the fracture pattern of the composite, differing from either pure cement pastes (one major crack) or other high-ductility cement-based materials, such as engineered cementitious composites (ECC, multi-cracks). It indicates a distinct toughening mechanism. In ECC, the added PVA fibers can bridge the micro-cracks to avoid the rapid propagation and convergence of one major crack. The orientation of PVA fibers is random, and the microstructure of ECC can be considered homogenous, which leads to a multi-cracking failure mode⁴⁹. However, this cement-hydrogel composite has a layered microstructure, where PVA hydrogels within neighboring cement lamellae can prevent or deflect the cracks on the interfaces due to their low elastic modulus and high cohesiveness. Therefore, the final fracture mode of this composite is as similar as that of nacre^{24,25}, with a limited number of major cracks. More details with regards to the toughening mechanism can be referred

to the finite element analysis in Section 2.2.

Reference:

[24] Verho, T., Karppinen, P., Gröschel, A. H. & Ikkala, O. *Advanced Science*, 2018, 5, 1700635.

[25] Gim, J. Schnitzer, N, M Otter, L, et al. *Nature Communications*, 2019, 10, 4822.

[49] Li, V. C. & Kanda, T. *Journal of Materials in Civil Engineering*, 1998, 10, 66–69.

3 - The fabrication process of the cement-hydrogel composite is complex, with an ice-template process, in situ hydration and then vacuum pumping, and 2~3 freezing-thawing cycles. Can this be applied in practical large-scale construction scenarios? What should be improved in the fabrication process to make the future applications easier and more convenient?

Response:

The industrialization of construction is becoming an increasingly popular trend, and the adoption of pipelined factory production methods is enabling the large-scale production of the cement-hydrogel composite material while simultaneously reducing costs.

In addition, future developments may allow for the application of this method to larger volumes by adjusting the freezing environment and filling method. The freezing environment is essential for constructing an orderly arrangement of the cement matrix, while vacuum negative pressure impregnation is used to fill the pore structure with hydrogel solution.

To make it clear, we have added the following discussion at Page 16, line 375-380.

Revision:

With a future development trend of construction industrialization, the assembly line production in factories can make the large-scale preparation of this cement-based composite possible and reduce the costs. Therefore, we believe the composite may promote a significant revolution of building materials, which can be widely applied in seismic high-rise or energy-conservation buildings, and long-span bridges.

4 - In section 4.7, some key parameters are missing in the Finite Element Method Simulations. Except for the geometry parameters of the model, the authors should also provide the mechanical modulus of the cement and hydrogel, respectively. Furthermore, how do they deal with the interface cohesiveness in the model of the cement-hydrogel composite?

Response:

We understand the importance of providing complete and transparent information and we have made necessary additions to our revised manuscript, as shown in Section 4.7 (Page 22, Line 546-552).

Revision:

The elastic modulus and Poisson's ratios of the cement matrix were determined to be 26.8 GPa and 0.2, respectively. For the hydrogel, the corresponding values were 0.24 MPa and 0.48. The zero-thickness cohesive elements were used to simulate the fracture properties between the cement and the hydrogel. The cohesive elements obeyed the Maxs damage model and was set to be removed when the displacement reached 0.0005 mm, which was expressed as a fracture process between the cells, macroscopically.

5 - The conclusion section is too lengthy and may be difficult for readers to quickly understand the main points. To improve clarity, please revise the conclusion section by including only the most significant conclusions achieved in the investigation. Any detailed results or discussions should be moved to the results and discussion section.

Response:

As required, we have reorganized the conclusion section to be brief, as shown in Page 15-16, Line 349-380.

Revision:

A biomimetic cement-based composite has been fabricated by a two-step method. Firstly, a simplified ice-template mixing is utilized to produce an ordered platelet-like skeleton of cement pastes, with unidirectional pores aligned in the interlayer. Subsequently, PVA hydrogels are in situ filled into the unidirectional pores. By constructing such a layered microstructure of highly aligned cement pastes interspaced with PVA hydrogels, we realize a toughness improvement of over 175 times as high as pure cement paste. To the best of our knowledge, it is the largest increase in the toughness of cement-based materials that has been reported, 1~2 orders of magnitude higher than that of the most commonly used fiber toughening measures. In the meanwhile, the flexural strength of the pure cement is also improved by over 2 times.

Furthermore, the toughness enhancement mechanism of the cement-hydrogel composite has also been interpreted. The bonding strength of cement-hydrogel interface is close to the tensile strength of PVA hydrogel itself, which indicates super high affinities between the hard and soft phase. The excellent interface compatibility results from the stable adsorption of PVA molecules on the cement matrix. Molecular dynamics simulations suggest as the tensile stress is applied, the PVA hydrogel can be fully stretched and deformed before the interface de-bonding, which significantly improves the ductility of the cement-hydrogel composite. Besides, the FEM simulations implies that during the progressive interface failure process, this unique brick-mortar structure leads to distinct micro-crack deflections near the crack tip, also accompanied by interface sliding and platelets pull-out. Those mechanisms can effectively avoid the stress concentration and dissipate huge energy, which explains the large ductility increase in the flexural stress-strain curves.

In addition to the toughening and strengthening effect, the cement-hydrogel composite

is further accompanied with multi-functionalities, including low density (around 2/3 of the pure cement paste), thermal insulation (one order of magnitude lower than that of the pure cement), and self-healing. With a future development trend of construction industrialization, the assembly line production in factories can make the large-scale preparation of this cement-based composite possible and reduce the costs. Therefore, we believe the composite may promote a significant revolution of building materials, which can be widely applied in seismic high-rise or energy-conservation buildings, and long-span bridges.

REVIEWERS' COMMENTS

Reviewer #3 (Remarks to the Author):

The authors has satisfied all my concerns.

Reviewer #4 (Remarks to the Author):

The author has made more appropriate answers and revisions to the questions raised, and recommends publication

Reviewer #3

The authors has satisfied all my concerns.

Response:

We appreciate the comments from the reviewer #3, which are very valuable to improve the quality of our manuscript.

Reviewer #4

The author has made more appropriate answers and revisions to the questions raised, and recommends publication.

Response:

We thank the reviewer for carefully reading the manuscript and for providing positive and very valuable feedback.